# Formation, Diffusion and Simulation of Green Production Socialized Service Network for Smallholder Farmers Based on SEIRS Model

**Sishu Zhou and Hong Chen ***

College of Economics and Management, Northeast Forestry University, Harbin 150040, China;
sishuzhou@nefu.edu.cn
* Correspondence: chenhong@nefu.edu.cn

**Abstract:** (1) Background: The spread of agricultural green production technologies and systems among small farmers is affected by multiple factors such as subjectivity and objectivity. (2) Methods: Based on the marketability of agricultural green production socialization services (AGPSSs), this paper constructs a SEIRS model of infectious disease dynamics, taking the AGPSS of "MAP Sinochem Modern Agriculture" in Tianshan Town, Arhorchin Banner as an example. (3) Results: This study uses Python to simulate the process of forming a network of AGPSS for small farmers and analyzes the law of information dissemination among farmers. (4) Conclusions: This paper explores how multiple factors such as service quality, external environment, farmers' willingness to decide, government guidance and the responsibility of service subjects play roles in the formation and diffusion of an AGPSS network so as to improve the quality and level of AGPSS provided by enterprises.

**Keywords:** socialized services for agricultural green production; SEIRS model; MAP Sinochem Modern Agriculture; dynamic simulation





## 1. Introduction

Under the basic national conditions of small farmers in large countries [1], the socialized service of agricultural green production [2–5], with the goals of protecting the environment, realizing the green transformation of agriculture and promoting the high-quality development of agriculture, is an inevitable product when agricultural production develops to a certain stage [6,7]. By providing green production technologies, low-carbon production factors and green production systems, the factor input structure of small farmers is changed, and the purpose of "expanding the scale of agricultural green production, improving the efficiency of agricultural green production, and increasing farmers' income" is realized. In the new stage of realizing a strong agricultural country, local governments and enterprises need to accelerating the development of agricultural green production socialized service (AGPSS) and constantly improving service capability and level. Solving the problems that many small farmers say "can't do it", "can't do it well" and "not cost-effective" is the research difficulty and focus of academic and practical departments.

At present, the research on agricultural socialization services and farmers' green production behaviors is mostly concentrated in three perspectives. First, from the perspective of farmers' needs, this paper studies the impacts of farmers' characteristics and attributes on their choice of social services. For example, Zhang Xiaomin and Jiang Changyun [8] found that the differentiation of farmers is the key factor affecting farmers' willingness to choose productive services. Farmers who are mainly agricultural-oriented and have poor family endowments are more willing to choose productive services. Liu Wenxia and Du Zhixiong [9], based on the monitoring data of planting family farms across the country, studied that family farms with different types and characteristics need different social services. Second, from the perspective of social service supply, this paper studies how to

improve the supply subjects, supply modes and the paths of social agricultural production services. For example, Guan Shan [10] believes that in order to improve the mechanism for interest linkage between service subjects and small farmers, it is necessary to take the market as the basis and give a guiding role to county governments; Xing Meihua [11] found that self-employed households in Hubei Province are the main providers of agricultural machinery, drainage and irrigation services. Based on the decision-making model of agricultural machinery service introduction, Dong Huan and Guo Xiaoming [12] found that the introduction of agricultural machinery services may lead to the regression of the business model to the extensive type. Thirdly, from the perspective of the balance between social service providers and demand subjects, this paper studies how "information, risk, cost and policy" affect the formation and development of a social service system for agricultural green production. For example, Zhuang Lijuan et al. [13]. found that the source of service information is an important influencing factor, and members of cooperatives tend to choose the purchasing and selling services of agricultural supplies. In the case of information asymmetry, the service price and surrounding people's behaviors will significantly affect the decision of farmers to sign services [14]; the short-term service purchased by farmers out of prudence will affect the service quality and the interests of both service subjects [15]; and the service subject's reduction in service quality and farmers' excessive supervision will result in bilateral moral hazard [16]. The cooperation between service subjects and family farms not only achieves a high coverage rate of agricultural productive service land, but also collects state subsidies, which fails to fundamentally solve the needs of small farmers' land trusteeship [17]. Although foreign enterprises can break through the technical and financial restrictions that localized service subjects are prone to, they are easily excluded in the social acquaintance network [18] full of local human feelings [19]. Therefore, it is necessary to explore the formation and diffusion mechanism of the social service network for agricultural green production from multiple perspectives, which is based on the market economy, respects the independent management status of farmers and ensures the sustainable operation of service providers.

Established in the early 20th century, the infectious disease model initially used mathematical methods to explain the transmission process of malaria between mosquitoes and people, and was later widely used to explore the transmission mechanism of the virus. It can quantify the short-term or long-term impact of various influencing factors through mathematical model derivation, simulate the change trend and effectively discuss the effect of various policies and measures on preventing the spread of the virus [20]. Since the spread process of new technologies and new institutions between economic and social systems is similar to the spread of infectious diseases within the same population or between different populations, the transmission law of new technologies and new institutions can be revealed according to the infectious disease model, and the infectious disease dynamics model has been applied to explore the transmission law and path of new technologies and new institutions. Zhang Zhanjun [21] used the SIR (Susceptible Infected Recovered) model to describe the endogenous risk transmission process of agricultural product supply chain reconstruction, and simulated the evolution trend under effective management and control. Cui Jindong et al. [22] modified the SEIR (Susceptible Exposed Infected Recovered) model to summarize the law of Internet topic information transmission; Zhou Fuli and Ye Zhengmei [23] built the SIR model to simulate the transmission process of the negative word of mouth of automobile enterprises. In addition, there are relevant studies on the dissemination of positive content such as technology, knowledge and crowdfunding. Ma Liping et al. [24] summarized the laws and mechanisms of disruptive technology diffusion based on the SEIR model, and offered suggestions for government guidance and cultivation. Yang Xianghao et al. [25] built a tacit knowledge transmission SIR model by considering the forgetting mechanism to provide a theoretical reference for planning the training cycle of employees; Cao Guang and Shen Lining [26] combined the SEIR model to analyze individual characteristics and behavioral decisions in the communication

of medical crowdfunding projects, providing support for improving the rationality and fairness of medical financing.

The formation process of an AGPSS network can be seen as the transmission and replication process of green innovation technologies and systems among farmers, which is the meaning of "service information" below. Moreover, farmers have a time lag from understanding green innovation technologies and systems of social service to signing a contract. In this process, the quality of social services and the government's policy guidance will affect the penetration and dissemination of green innovation technologies and systems among farmers. In this paper, an AGPSS network is constructed exploratorily by using an infectious disease dynamics model; the formation and diffusion process of the service network is depicted with smallholder farmers as the service objects. The MAP green production service model [27] of Tianshan Town of the Arhorchin Banner of Sinochem Group is taken as an example for simulation to explore the diffusion process, mechanism and conditions of an AGPSS network, which covers the whole process, provides comprehensive services, operates efficiently and provides comprehensive support, relying on the leading enterprises. This article is organized as follows: Section 2 describes the construction and threshold analysis of the system dynamics model. Section 3 demonstrates and discusses the simulation results. Section 4 makes relevant policy recommendations. Section 5 presents some conclusions.

## 2. Materials and Methods

### 2.1. Formation Mechanism of Service Network

There is asymmetric information between the providers of innovative technologies and institutional services for agricultural green production and the demand subjects [28,29]. Farmers have low ability to supervise the service quality provided by the service subjects, and the acceptance of the service by farmers in the initial stage is low, too. Only through extensive dissemination and demonstration can the service be accepted and disseminated by more farmers. If a village or contiguous villages and towns are regarded as topological structures, the AGPSS network is composed of farmers as nodes. According to the "able man effect" [30] of rural society, the degree of nodes of village cadres and farmers is high, and the distribution of degrees is uneven. Therefore, the AGPSS network is a scale-free network [31] that obeys the power law distribution. Information dissemination in scale-free network is a "one-to-many" mode, that is, social service providers such as cooperatives or leading enterprises serve farmers in the region, and some farmers spread the information they know to a number of acquaintances with the same needs, resulting in a willingness to sign contracts as potential customers. After the publicity of service providers and the research and judgment of farmers, some farmers sign contracts. Some farmers wait and see and then continue to cycle the above process until a stable social service demand group for agricultural green production is formed.

This paper gives node interaction rules combined with the bottom-up modeling method of the SIERS model. According to farmers' acceptance of green innovation technologies and systems, it can be considered that in the formation process of an AGPSS network, farmers are divided into four types: target customer S (all farmers within the scope of social service), potential customer E (farmers with contract intention), contract customer I (real contract farmer) and non-service farmer R. Among them, group R has three sources. First, the target customers do not know the service information and directly reject the service. Second, the potential customers refuse to sign the contract. Third, the signed customers refuse to renew the contract. After the first round of contracts expires, the four types of farmers have related characteristics of transformation according to changes in the social service quality, agricultural policy, production environment and the characteristics of the farmers themselves. This process is the formation and diffusion process of an AGPSS network. According to the relevant regulations and policies of Chinese agricultural green production and the practice of agricultural green production socialization services, this paper assumes that the number of farmers served by AGPSSs remains unchanged for the

purpose of research. $S_k(t)$, $E_k(t)$, $I_k(t)$ and $R_k(t)$, respectively, represent the k state of four types of nodes. The number of farmers in time, t, meet $S_k(t) + E_k(t) + I_k(t) + R_k(t) = N$, and then $s_k(t) = S_k(t)/N$, $e_k(t) = E_k(t)/N$, $i_k(t) = I_k(t)/N$, $r_k(t) = R_k(t)/N$ [32]. The information dissemination model of an AGPSS is shown in Figure 1.

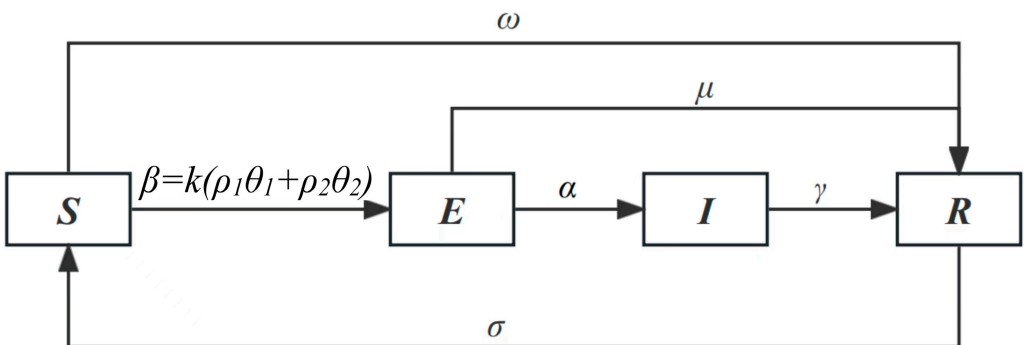

**Figure 1.** AGPSS network based on SEIRS model.

*2.2. Model Parameter Analysis*

$\beta$ is the contagiousness, representing the probability that farmers in the region know the AGPSS information and have signing intention. $\beta$ is determined by the quality and promoting intensity of AGPSS and is affected by farmers' ability to study and judge external information and the external environment. The better the service quality and the effectiveness of government policy guidance, the stronger the farmers' abilities to accept green technology, and the stronger the responsibility of environmental protection, the higher the value of $\beta$. Because of the change in network topology, the expression of contagion in scale-free network is different from that in small-world network. According to the equilibrium field theory of epidemic transmission in heterogeneous networks, proposed by Moreno et al. [33], the contagiousness of the social service network for agricultural green production is set. $\beta = k(\rho_1\theta_1 + \rho_2\theta_2)$, $\theta_1 = \sum kp(k)E_k(t)/\langle k \rangle$, $\theta_2 = \sum kp(k)I_k(t)/\langle k \rangle$, where $\langle k \rangle$ is the degree distribution and $p(k) = 2m^2k^{-3}$. $m$ represents the number of nodes connected to the original network at each time step. The corresponding meanings are shown in Table 1. $\beta = k(\rho_1\theta_1 + \rho_2\theta_2)$ indicates that the process of infectious disease transmission in a scale-free network must be followed as $S \rightarrow E \rightarrow I$.

**Table 1.** Parameters and meanings of information dissemination model of AGPSS.

| Parameters | Infectious Disease Model Implications | AGPSS Network Diffusion Model Implications |
|---|---|---|
| $\beta$ | The conversion rate of susceptible nodes to latent nodes | The probability that farmers understand the service and have intention to sign contract |
| $\rho_1$ | The conversion rate of susceptible nodes being infected into latent nodes | The probability of signing intention after contact and communication between farmers who do not know and farmers who have signing intention |
| $\rho_2$ | The conversion rate of susceptible state nodes being infected to infected state nodes | The probability of signing intention after contact and communication between farmers who do not know and farmers who have signed a contract |
| $\theta_1$ | The probability that a node in the latent state at time $t$ is connected to a random edge | The probability that a farmer with contract intention communicates with any farmer at time $t$ in a scale-free network |
| $\theta_2$ | The probability that a node in the infected state at time $t$ is connected to a random edge | The probability that a contracted farmer at time $t$ communicates with any farmer in a scale-free network |

**Table 1.** *Cont.*

| Parameters | Infectious Disease Model Implications | AGPSS Network Diffusion Model Implications |
|---|---|---|
| $\alpha$ | The conversion rate of latent node to infected node | The probability of famers understanding and accepting the service contract |
| $\mu$ | The conversion rate of latent node to immune node | The probability of famers understanding but refusing the service contract |
| $\gamma$ | The conversion rate of infected node to immune node | The probability that farmers who are using social services will not renew their contracts |
| $\sigma$ | The conversion rate of immune node to susceptible node | The probability of farmers who had refused to sign up to re-accept services |
| $\omega$ | The direct conversion rate from susceptible node to immune node | The probability of direct refusal of service by farmers who do not know the service information |

$\alpha$ is the infection rate, which represents the rate of signing the AGPSS contracts. The value of $\alpha$ mainly depends on the subjective will of potential customers. The higher the demand for services, the lower the cost of green innovation technology, and the stronger the environmental regulation, the easier it is to encourage farmers to sign up and the higher the value of $\alpha$. On the other hand, the better the reputation of the social service supplier and the stronger its ability to resist negative public opinion, the more it can enhance the confidence of potential customers to convert into contracted customers, and correspondingly the $\alpha$ value is higher.

$\gamma$ is the cure rate, representing the probability that farmers who are adopting AGPSS will refuse to renew their contracts. The value of $\gamma$ mainly depends on the duration of a single contract and the ability of both supply and demand to withstand the impact of agricultural production risks. Too long a single contract period is not conducive to the acceptance of farmers, while too short a single contract period is not only conducive to the sustainable use of land, but also harms the interests of service subjects. Agricultural production risks affect farmers' income, and sudden drops in output and income will inevitably affect farmers' contract renewal behavior, resulting in a relatively high $\gamma$. The service subjects whose responsibility is the green transformation of agricultural production can obtain government support and subsidies to offset the contract cost of farmers. Their ability to resist risks will be relatively strong, which is conducive to the dissemination of service information, and they will attract more loyal farmers and further promote the coverage of AGPSS.

$\sigma$ is the immune degradation rate, which represents the probability that farmers who do not renew their contracts or have never signed contracts are informed These farmers return to being target customers (S) and eventually transform into potential customers (E), contract customers (I) or non-participating service farmers (R). The value of $\sigma$ mainly depends on the green production technology and institutional renewal ability of the service supplier. The faster the feedback response to the opinions of farmers, the larger the value of $\sigma$. The incentive degree of the government to the service suppliers affects the cost and realization path of the social service suppliers to promote the green innovation technology, and also affects the value of $\sigma$.

$\mu$ is the latent node immunization rate, which represents the probability that the interested farmers will refuse to sign the contract. $\mu$ and $\alpha$ are the relationship between the ebb and flow. The more high-intensity negative information such as increased cost, decreased farmers' income, and the declined enterprises' credibility of AGPSS, the more farmers will refuse to sign contracts, which leads to the increased $\mu$.

$\omega$ is the direct immunization rate, representing the probability of farmers rejecting social services when they do not understand the agricultural green technology and system. The value of $\omega$ mainly depends on the strength of signals such as the quality and level of AGPSS, the sensitivity of target customers to information and the dissemination envi-

ronment of green service information. If the social service supply capacity of agricultural green production is weak, the acceptance of farmers is not sensitive, the communication channel of information is not smooth, the contact between potential customers and target customers is less, and the communication frequency is low, these will lead to the increase in $\omega$.

## 2.3. Model Construction and Threshold Analysis

The differential equations of the system are established according to the system dynamics modeling method [34], as shown in Equation (1):

$$
\begin{cases}
\frac{ds_k(t)}{dt} = -k(\rho_1\theta_1 + \rho_2\theta_2)s_k(t) - \omega s_k(t) + \sigma r_k(t) \\
\frac{de_k(t)}{dt} = k(\rho_1\theta_1 + \rho_2\theta_2)s_k(t) - \alpha e_k(t) - \mu e_k(t) \\
\frac{di_k(t)}{dt} = \alpha e_k(t) - \gamma i_k(t) \\
\frac{dr_k(t)}{dt} = \omega s_k(t) + \mu e_k(t) + \gamma i_k(t) - \sigma r_k(t)
\end{cases}
\tag{1}
$$

If the four equations in system (1) are equal to 0, the zero-propagation equilibrium point $(0,0,0,0)$ can be obtained, which is practically meaningless. The equilibrium point can be obtained according to the property of the non-zero equilibrium point, $\left(\frac{(\alpha+\mu)\gamma}{k\rho\alpha}i^*, \frac{\gamma}{\alpha}i^*, i^*, \frac{(\omega\alpha+\omega\mu+k\rho\mu+k\rho\alpha)\gamma}{k\rho\alpha(1+\sigma)}i^*\right)$, where $i^* = \frac{k\rho\alpha\sigma}{k\rho\alpha\sigma + \gamma[\sigma(\alpha+\mu)+k\rho\sigma+(k\rho+\omega)(\alpha+\mu)]}$.

When $t \to \infty$, $\rho(\infty) = \lim\limits_{t\to\infty}\rho(t)$ can be obtained, which means

$$
i^* = i(\infty) = \frac{k\rho(\infty)\alpha\sigma}{k\rho(\infty)[\alpha\sigma + \gamma(\sigma+\alpha+\mu)] + \gamma[(\alpha+\mu)(\sigma+\omega)]}
\tag{2}
$$

Because

$$
\begin{aligned}
\rho(\infty) &= \rho_1\theta_1 + \rho_2\theta_2 \\
&= \rho_1\frac{kp(k)E_k(\infty)}{\langle k\rangle} + \rho_2\frac{kp(k)I_k(\infty)}{\langle k\rangle} \\
&= \frac{kp(k)}{\langle k\rangle}\left(\rho_1\frac{\gamma}{\alpha} + \rho_2\right)I_k(\infty)
\end{aligned}
\tag{3}
$$

When we connect Formulas (2) and (3), we obtain the following:

$$
\rho(\infty) = \frac{1}{\langle k\rangle}\left(\rho_1\frac{\gamma}{\alpha} + \rho_2\right)\frac{k^2\rho(\infty)\alpha\sigma}{k\rho(\infty)[\alpha\sigma + \gamma(\sigma+\alpha+\mu)] + \gamma[(\alpha+\mu)(\sigma+\omega)]}
\tag{4}
$$

When $\rho(\infty) = 0$, Equation (4) holds. It is its trivial solution. If Formula (4), $f(\rho(\infty))$, is assumed to be continuously differentiable, then it is a monotonically increasing convex function. The premise for the existence of non-trivial solution which means if the existence of $0 < \rho(\infty) < 1$ makes it possible to assume that $f'\big|_{\rho(\infty)=0} > 1$ is

$$
f'\big|_{\rho(\infty)=0} = \frac{\langle k^2\rangle}{\langle k\rangle}\left(\rho_1\frac{\gamma}{\alpha} + \rho_2\right)\frac{\sigma\alpha}{\gamma[(\alpha+\mu)(\sigma+\omega)]} > 1
\tag{5}
$$

Remember the basic reproduction number [35], $R_0 = \frac{\langle k^2\rangle}{\langle k\rangle}\left(\rho_1\frac{\gamma}{\alpha} + \rho_2\right)\frac{\alpha\sigma}{\gamma[(\alpha+\mu)(\sigma+\omega)]}$, which is the threshold at which the service information can be widely propagated in the system. When $R_0 < 1$, which is $i < 0$, there is no non-zero equilibrium point in the system, and the dissemination of social service information for agricultural green production will gradually weaken until it disappears. When $R_0 = 1$, which is $i = 0$, the system has a balance point. When $R_0 > 1$, which is $i > 0$, there is a unique non-zero equilibrium point in the system. The scale of social service supply for agricultural green production will be expanded, and the system converges will go to the equilibrium point. According to the threshold expression, the following conclusions can be obtained:

Conclusion 1: On the premise that the other factors remain unchanged, the extension service threshold ($\langle k^2 \rangle$, $\rho_1$, $\rho_2$) will increase with the increase in the probability of service information exchange among farmers, which is conducive to the expansive supply of AGPSS and the diffusion of agricultural green production technology.

Conclusion 2: On the premise that the other factors remain unchanged, when $\mu > \gamma$, the propagation threshold will increase with the rise of $\alpha$ and will be closer to or more than 1, which is more conducive to diffusion. When $\mu < \gamma$, the increase will reduce the propagation threshold, which is not conducive to information dissemination. The path proof is as follows:

$$
\begin{aligned}
\frac{dR_0}{d\alpha} &= \frac{d\left[\frac{\langle k^2 \rangle}{\langle k \rangle}\left(\rho_1 \frac{\gamma}{\alpha} + \rho_2\right)\frac{\alpha\sigma}{\gamma(\alpha+\mu)(\sigma+\omega)}\right]}{d\alpha} \\
&= \frac{d\left[\frac{\langle k^2 \rangle}{\langle k \rangle}\left(\rho_1 \frac{\gamma}{\alpha} + \rho_2\right)\right]}{d\alpha}\frac{\alpha\sigma}{\gamma(\alpha+\mu)(\sigma+\omega)} + \frac{\langle k^2 \rangle}{\langle k \rangle}\left(\rho_1 \frac{\gamma}{\alpha} + \rho_2\right)\frac{d\left[\frac{\alpha\sigma}{\gamma(\alpha+\mu)(\sigma+\omega)}\right]}{d\alpha} \\
&= \frac{\langle k^2 \rangle}{\langle k \rangle}\left(-\rho_1 \frac{\gamma}{\alpha^2}\right)\frac{\alpha\sigma}{\gamma(\alpha+\mu)(\sigma+\omega)} + \frac{\langle k^2 \rangle}{\langle k \rangle}\left(\rho_1 \frac{\gamma}{\alpha} + \rho_2\right)\frac{\sigma\gamma(\alpha+\mu)(\sigma+\omega) - \alpha\sigma\gamma(\sigma+\omega)}{[\gamma(\alpha+\mu)(\sigma+\omega)]^2} \\
&= \frac{\langle k^2 \rangle}{\langle k \rangle}\left[\rho_1 \frac{\gamma}{\alpha^2}\frac{-\alpha\sigma}{\gamma(\alpha+\mu)(\sigma+\omega)} + \left(\rho_1 \frac{\gamma}{\alpha} + \rho_2\right)\frac{\sigma\mu}{\gamma(\alpha+\mu)^2(\sigma+\omega)}\right] \\
&= \frac{\langle k^2 \rangle}{\langle k \rangle}\frac{-\sigma\gamma\rho_1 + \sigma\mu\rho_2}{\gamma(\alpha+\mu)^2(\sigma+\omega)}
\end{aligned}
\tag{6}
$$

As can be seen from the above results, this paper assume that the target customers are equally affected by potential customers and contracted customers ($\rho_1 = \rho_2$). Because the service subject provides timely feedback on the opinions of farmers and updates the service content, the probability of potential customers refusing to sign contracts is greater than the probability of contracted customers refusing to renew their contracts ($\mu > \gamma$); then, $\frac{dR_0}{d\alpha} > 0$. This means the transmission threshold increases with the increase in the signing rate, which is conducive to the transmission of service information. Since the service subject is backed by large enterprises and the service pricing is reasonable, the cost of the farmers' choice to produce social services is lower than the other decisions. So, the probability of potential customers refusing to sign contracts is smaller than the probability of contract customers refusing to renew their contracts ($\mu < \gamma$); then, $\frac{dR_0}{d\alpha} < 0$. This means the transmission threshold decreases with the increase in the signing rate. Increasing the transmission ratio of potential customers will lower the threshold for AGPSS information to spread. This is not conducive to the spread of AGPSS network.

## 3. Results and Discussion

### 3.1. Case of "MAP Modern Agriculture"

Sinochem China is the largest central enterprise integrating fertilizers, seeds and pesticides in the country, with a leading production and comprehensive service system. It has built a new social service model of "leading enterprises + cooperatives + farmers" for agricultural green production. The Modern Agriculture Platform (MAP), a social green production service, is a tracking service for the whole industrial chain of agricultural green production, including seed breeding, whole process quality assurance, soil improvement, agricultural machinery service, order service, financial service and smart agriculture service. Since 2017, the MAP social service model has been organically integrated with rural revitalization and industrial poverty alleviation projects, which has promoted the agricultural green transformation of the Arhorchin Banner of Nei Mongolia. This paper takes the MAP project of China as an example to simulate the AGPSS network in order to explore the factors that affect the formation and diffusion of the network.

Located in the central part of the Nei Mongol Province, under the jurisdiction of Chifeng City, the Arhorchin Banner has a temperate continental climate, sitting in the

eastern foothills of the southern section of the Daxing'an Ridge. The Arhorchin Banner has a total population of 300,000, with a total land area of 14,277 km$^2$, of which 1033 km$^2$ is arable land, and the semi-arid climate environment makes it a typical agro-pastoral area. In 2019, the MAP project by Sinochem China has built six demonstration farms with areas of 11.287 km$^2$, and now, it has reached an area of 805 km$^2$. Through serving village collective cooperatives, large planters and prosperity leaders, driving smallholder farmers to improve planting technology, promote appropriate scale land management and cultivate new agricultural management subjects and new professional farmers, Sinochem China established a sustainable MAP pattern for agricultural service.

### 3.2. Analysis of the Influence of Time Dimension

This paper uses Python to simulate the dynamic changes of farmers' roles in the AGPSS network and explores the dynamic process of the formation and diffusion in combination with MPA cases. To express clearly, this paper defines lower letters such as "s", which represents the proportion of target customer S. The initial model setting is $s = 1 - e - i - r$, $i = 0.001$, $e = r = 0$. According to the six degrees of separation theory, $\langle k \rangle = 6$ is set. The other parameters are set as follows: $\beta = 0.8$, $\alpha = 0.3$, $\gamma = 0.01$, $\sigma = 0.15$, $\mu = 0.1$, $\omega = 0.04$.

As can be seen from Figure 2, the target customer S drops rapidly and becomes stable. The proportion of potential customer E rises first and then falls and becomes stable. The proportion of accepted customer I rises rapidly in the initial stage and gradually becomes stable. And the proportion of customer R, who does not renew the contract, rises first and then falls and becomes stable, but the range is much smaller than that of potential customers. Under the initial setting conditions, the service coverage rate is stable at about 80%~90%, which is in accordance with the actual situation of the MAP project of Sinochem. The MAP green production social service model first adheres to the principle of science, adopts the promotion mode of "pilot first, with points and areas" and establishes demonstration farms by unifying planting technology standards; agricultural supplies; agricultural machinery services; the comprehensive prevention and control of diseases, pests and grasses; and financial management so as to promote the development of the surrounding green agricultural industry. With the popularization of the pilot experience, the savings rate of farmers has increased year by year.

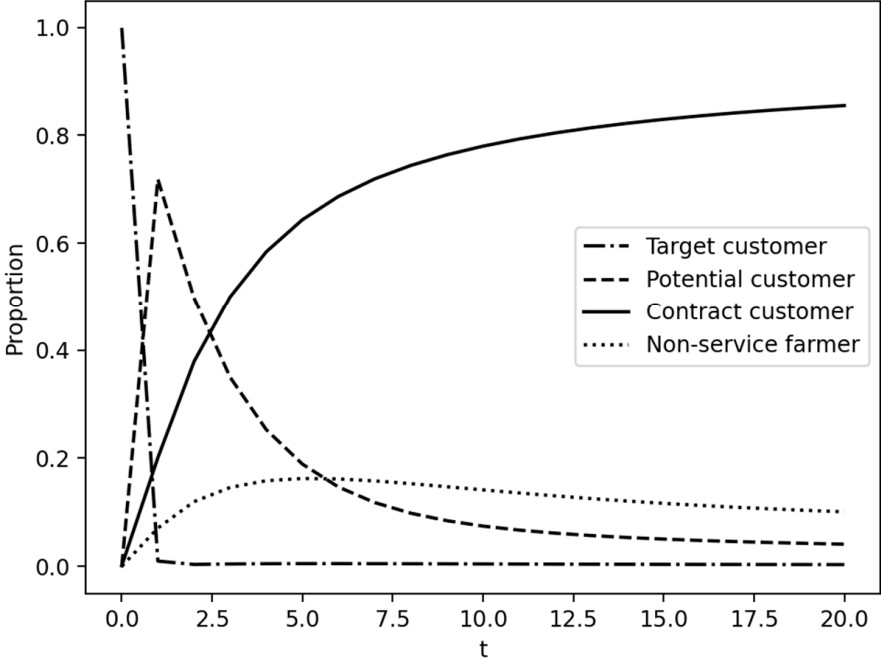

**Figure 2.** Simulation result of AGPSS network based on SEIRS model under initial setpoint.

### 3.3. Analysis of the Impact of the Change in Signing Rate

Farmers who understand service information as well as have the intention to sign contracts are the keys to expanding the scale of an AGPSS. When the direct immunization rate of latent persons is greater than that of infectious persons, the transmission threshold will increase with the increase in $\alpha$, which is conducive to the transmission of service information. The value of $\alpha$ is increased from 0.3 to 0.5, 0.7 and 0.9, respectively, and the change in accepted service customer I is shown in Figure 3. It can be seen that the larger the $\alpha$ value, the higher the stable value of $i$ and the shorter the time to reach a steady state.

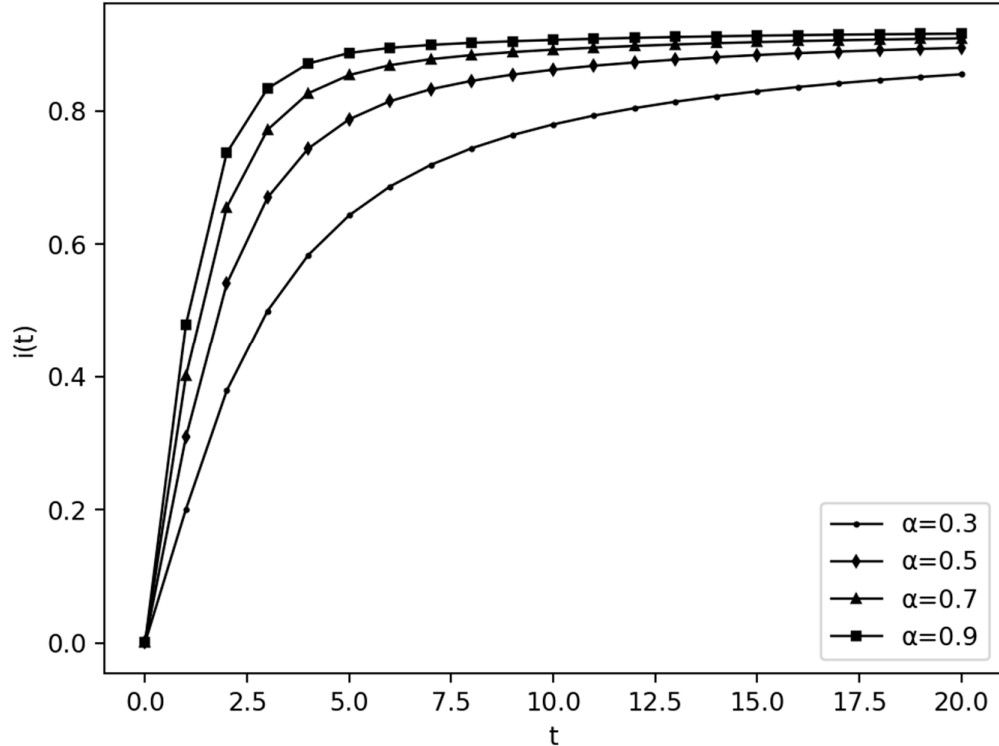

**Figure 3.** The effect of the signing rate on the number of customers in the I classification who have accepted the service.

The simulation results show that the increase in the ratio of potential customers to contract customers can effectively expand the dissemination scale of service information. The improvement in the signing rate depends on the subjective willingness of potential customers. If farmers subjectively recognize the social service of green production but do not sign, this means the identity of potential customers remains unchanged, and the paradox of "high satisfaction and low execution" will appear in the early stage of the promotion of social services of agricultural green production. If farmers can continue to feel the benefits of social services and recognize the MAP service model in communication, the signing rate will be improved.

### 3.4. Impact Analysis of Agricultural Production Risk

Given the frequency, unpredictability and impact intensity of agricultural production risk outbreaks, this study further simulates the impact of major risks on the formation and diffusion of an AGPSS network. According to the Model Text of Agricultural Production Trusteeship Service Contract issued by the General Office of the Ministry of Agriculture and Rural Affairs of China, the parties of the contract need to agree on the average crop yield per mu or the income per mu. If the agreed yield or income is not reached, the two parties will negotiate the compensation amount or specific compensation method. When natural disasters affect production and income, if service providers cannot pay or bear farmers'

losses in time, they will directly affect the farmers' decision to renew their contracts, leading to farmers' refusal to renew their contracts, which is manifested as a significant increase in the immune conversion rate $\gamma$ in the model. The value of $\gamma$ was increased from 0.01 to 0.3, 0.5 and 0.7, respectively, and the change in the quantity ratio of contracted farmer I is shown in Figure 4. It can be seen that the larger the $\gamma$ value, the faster $i$ reaches the peak and falls back to a stable state, but both the peak value and the stable value decrease with the increase in the $\gamma$ value. Due to the robustness of the scale-free network, there will still be a certain proportion of contracted farmers without affecting the income of most farmers.

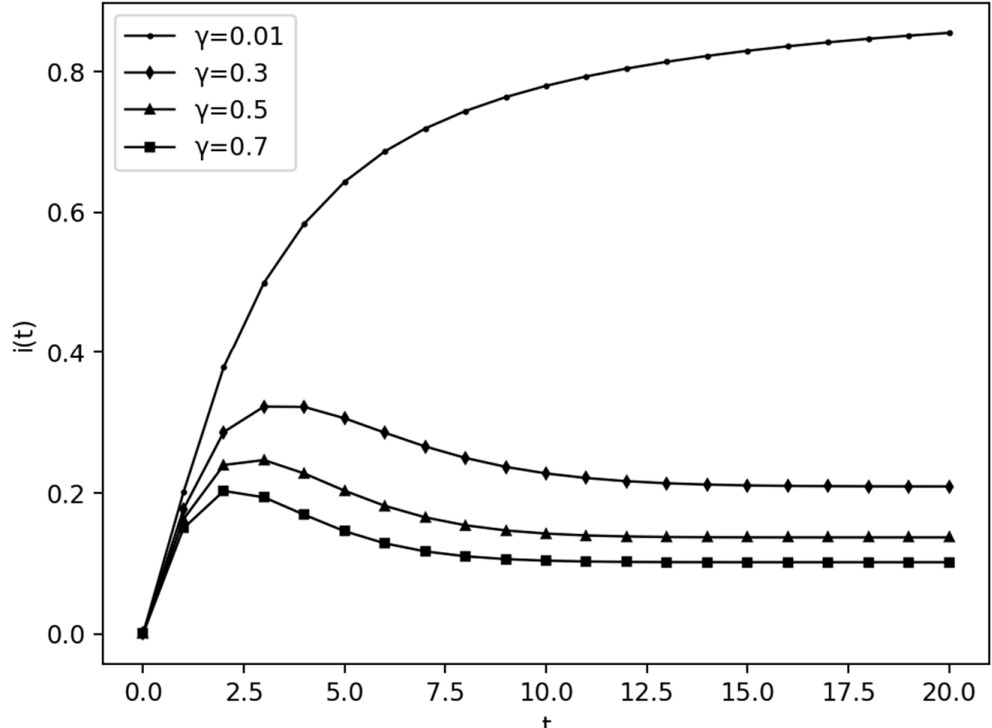

**Figure 4.** The impact of agricultural production risk on the proportion of farmers who are using AGPSS.

The simulation results show that an AGPSS has a production risk resistance effect and the robustness of a complex network. Figure 5 shows the impact of the agricultural production risk on the AGPSS network model. It can be seen that the steady-state values of potential customer E and non-participating farmer R increase with the increase in the $\gamma$ value. Compared with non-participating farmer R, the steady-state values are more sensitive, and the target customer S is basically not affected. Figure 6 shows that when $t = 2$, the $\gamma$ value changes from 0.01 to 0.7, and when $t = 3$, the $\gamma$ value changes again to 0.01. It can be seen that sudden agricultural production risks have immediate impacts on the service network of agricultural production, and the longer the time, the greater the impact. However, the impact can be reduced through agricultural insurance and service guarantee, and the service network can be repaired. The quantity ratio of contracted farmers in class I can still rise to about 80% and become stable. Based on the principle of improving land efficiency and benefiting farmers, the MAP AGPSS of Sinochem realizes the goal of "reducing fertilizer and medicine" under the premise of ensuring the safety of agricultural production. MAP makes planting plans for farmers, provides the whole process of agricultural machinery services, the whole process of plant nutrition services, the whole process of plant protection services and provides farmers with technical training.

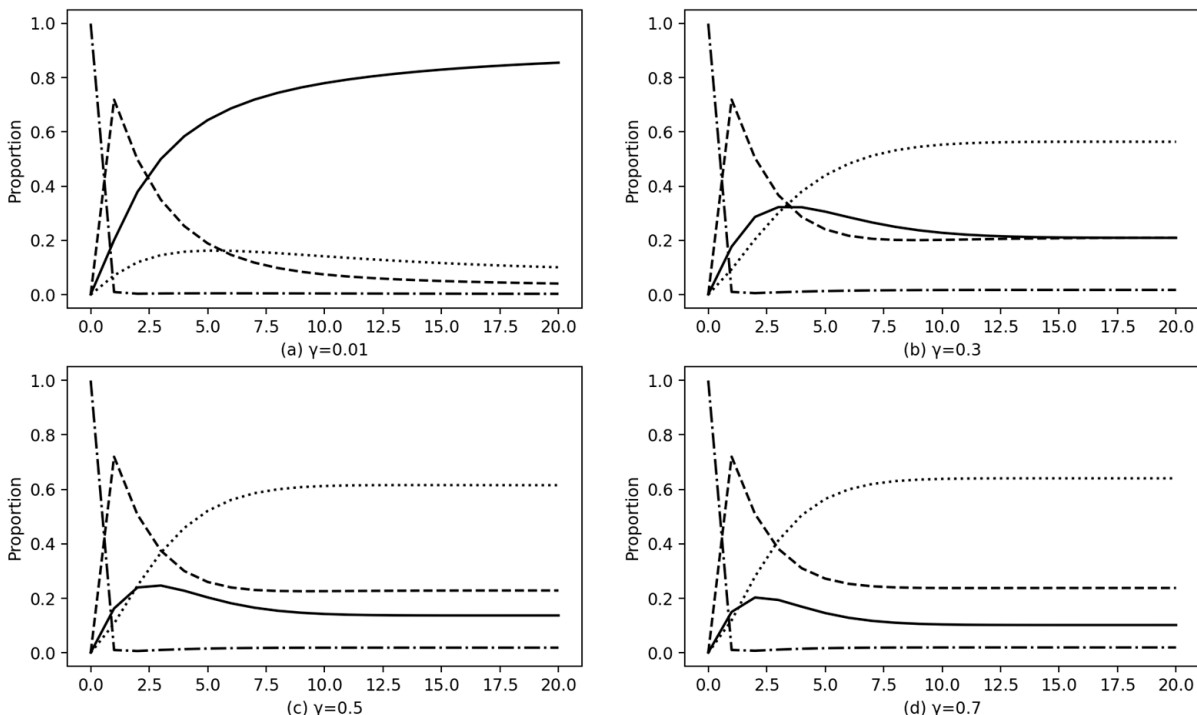

**Figure 5.** Influence of agricultural production risk on AGPSS network model. See Figure 1 for the explanations of the lines.

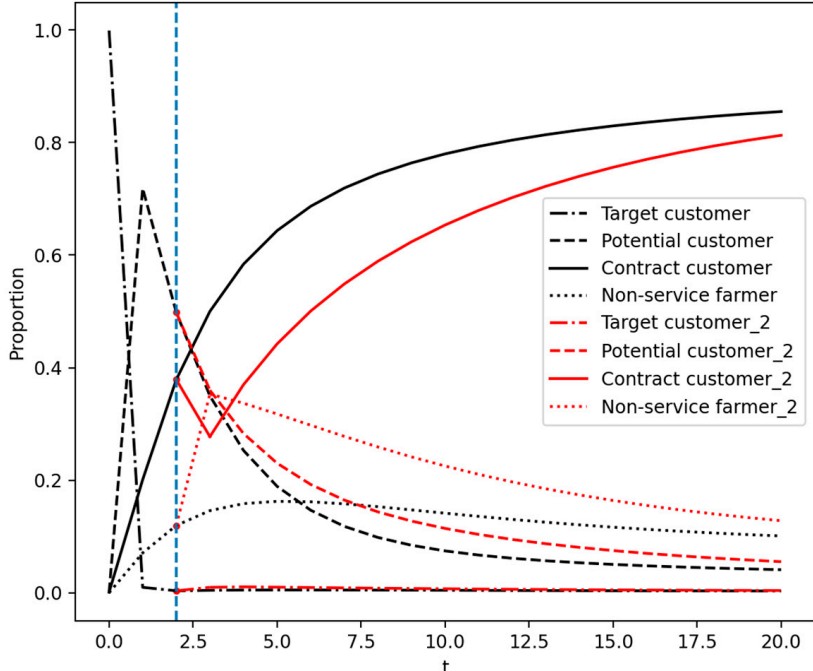

**Figure 6.** The impact of sudden agricultural production risk and resolution on the AGPSS network model. The blue line indicates t = 2.

### 3.5. Impact Analysis of Feedback Response and Subsidy Policy

The simulation results show that improving the service feedback speed can strengthen the service response, and a government policy adjustment can promote the formation and diffusion of a social service network for agricultural green production. The MAP AGPSS is a pattern with multi-subject participation and multi-objective integration. Government

supporting policies and financial support can influence the formation and diffusion of social service networks by influencing the behaviors of farmers and the interests of service providers. The timely resolution of the demands of local governments and service suppliers to farmers can improve the conversion rate of farmers who refuse to renew their visa to target customers for the first time, which is shown in the model as a significant increase in the immune degradation rate $\sigma$. The $\sigma$ value is increased from 0.15 to 0.3, 0.45 and 0.6, respectively, and the change in the class of accepted service customer I is shown in Figure 7. It can be seen that the change in $\sigma$ has basically no effect on the initial rise rate of $i$, but the increase in $\sigma$ will improve the steady-state value of $i$ and shorten the time to reach the steady state. The MAP green production socialization service takes serving small farmers as the main object of policy support, and focuses on solving the problem of the large-scale production of small farmers. The MAP sticks to the promotion of a service-led scale operation, breaks the constraints of small-scale decentralized operations and develops large-scale agricultural production. In addition, the local government provides some supporting policies. The high standard farmland construction of the government covers all the plots, and free deep loosening projects or social service subsidies of CNY 1500~2250 per hectare can cover all of the plots. The poverty alleviation fund of Sinochem invested CNY 10.8 million for the execution funds of the project, including the circulation funds needed to achieve the land scale, the entire production and operation costs and the construction of farmland water conservancy conditions.

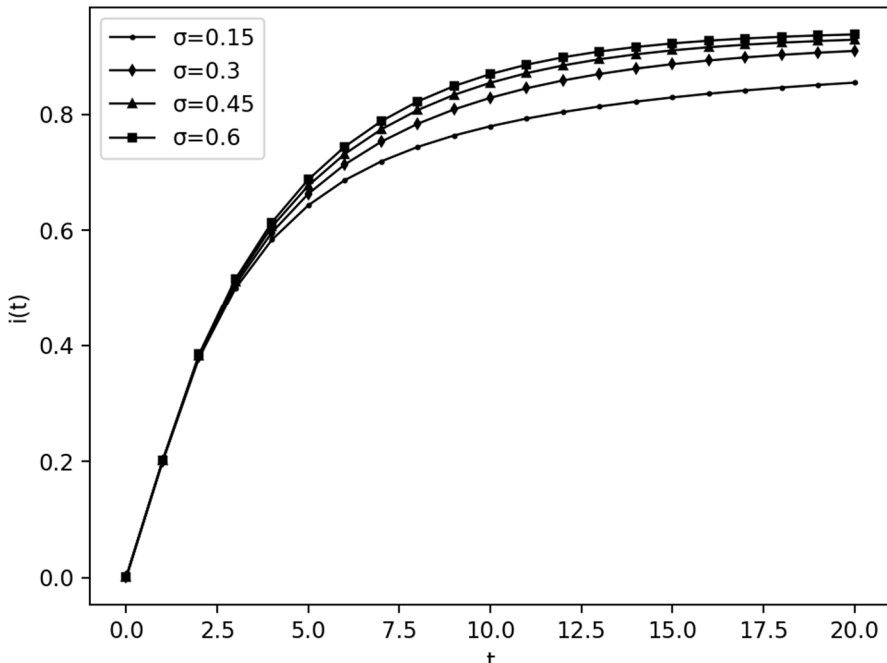

**Figure 7.** Influence of feedback response and subsidy policy on the number of accepted service customers in class I.

In summary, this paper believes that the SEIRS model can simulate the internal influence process of an AGPSS network, reveal the multi-dimensional factors that affect the coverage rate of an AGPSS and explain the formation and diffusion of an AGPSS network facing small farmers by leading enterprises.

## 4. Policy Implications

Based on the research of this paper, in order to accelerate the high-quality development of agricultural green transformation, meet the growing demand of small farmers for agricultural green social services and improve the level of AGPSSs, the following countermeasures and suggestions are proposed:

Firstly, enterprises that provide AGPSSs should have a deep understanding of the social environment of their rural acquaintances, effectively unite with local rural basic organizations and cooperatives and gradually expand the scale of services and broaden the field of services.

Secondly, service entities should improve the quality of AGPSSs and stabilize the signing rate of rural households. They should work hard to solve the small farmers' problems, such as those "who cannot do their jobs", "who cannot do their jobs well", "who are not cost-effective" and so on. The scope of services should be extended from production to prenatal, post-natal and other links, as well as financial and insurance services.

Thirdly, the government combined with the enterprises should innovate social service models for green agricultural production. We encourage the MAP modern service concept of "planting to farmers, taking farmers to work", building a consortium of "agricultural green production service +", providing "R&D, demonstration, delivery, improved varieties + improved methods" and other services throughout the whole process of agricultural green production.

Furthermore, digital intelligence enables the formation of an "agricultural green value chain co-creation and sharing platform" with the participation of various agricultural planting subjects and value chain partners.

## 5. Conclusions

This paper uses the SEIRS model of infectious disease dynamics to build an AGPSS network, applies Python to simulate the formation process of an AGPSS network and quantitatively explores the complex influence of multi-dimensional factors such as the information itself, external environment, farmers' willingness to make decisions, government guidance and service subjects' responsibility on the formation and diffusion process of an AGPSS network. The following conclusions were drawn:

Firstly, the SEIRS model of infectious disease dynamics can better solve the behavioral motivation of farmers to participate in the AGPSS in the social environment of rural acquaintances, and it is suitable for describing the formation and diffusion of an AGPSS network.

Secondly, an AGPSS network will be affected by the contact rate and signing rate among farmers. When the probability of potential customers refusing to sign contracts is greater than the probability of contract customers refusing to renew contracts, the transmission threshold will increase with the increase in the signing rate, speeding up the formation and spread of the service network.

Thirdly, the simulation results of the agricultural green production socialization service of "MAP Sinochem Modern Agriculture" in Tianshan Town of the Arhorchin Banner show that the time dimension, farmer signing rate, agricultural production risk, feedback response speed of service providers and adjustment of government subsidy policies all have impacts on the simulation results, which eventually reach equilibrium.

**Author Contributions:** Conceptualization, S.Z. and H.C.; writing—original draft preparation, S.Z.; supervision, H.C. All authors have read and agreed to the published version of the manuscript.

**Funding:** This research was supported by the National Social Science Foundation of China (Grant Reference: 22BJY089), the Special Fund for Basic Research Funds of Central Universities (Grant Reference: 2572022DE01), the Heilongjiang Provincial Social Science Foundation Project (Grant Reference: 21JYB149), the Heilongjiang Provincial University Think Tank Key Project (Grant Reference: ZKKF2022172), and the Key Research Project of Economic and Social Development of Heilongjiang Province (Grant Reference: 20209). The authors would like to thank the editors and anonymous reviewers for their insightful comments.

**Institutional Review Board Statement:** Not applicable.

**Data Availability Statement:** Not applicable.

**Conflicts of Interest:** The authors declare no conflict of interest.

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
