# Peer review of "Formation, Diffusion and Simulation of Green Production Socialized Service Network for Smallholder Farmers Based on SEIRS Model"

_agriculture, doi:10.3390/agriculture13101963_

Round 1

Reviewer 1 Report

The proposed paper offers an interesting reading of a relevant and topical issue.

The proposed approach is new and interesting, but the text is not always clear and the procedure used is not easy to follow and completely understand, as it is presented.  Important explanations seem missing: how contracts work, why that MAP was chosen and what the Python model is. The bibliography, practically only Chinese, should be supplemented.

A first important aspect is the absence of a clear description of the contracts (line 105) that farmers could sign. What exactly are they? What constraints and opportunities do they offer? For example, why when natural disasters come service providers should pay or bear farmers' losses (row 350)? Do the contracts also provide for insurance or risk mutualisation?

 A second point that should be improved is the exposition.  The text is not very fluent, and the reader gets lost at times. Acronyms should also be used to facilitate reading (e.g. Agricultural socialisation service ASS). Please check the terms used in the reports and review the layout of part 2.3 Model construction (lines 228-286 and lines 305-308), because the representation is difficult to follow. The formulae should be well separated from the text and numbered to make reading easier. It is also advisable to carefully review the iconographic parts, the axis indications (time is expressed how?) and the legends. About the legends, it would be better to refer to the wording used in Chapter 2.1 and not to those relating to the spread of an infection (see figure 2: instead of 'Susceptible' use 'S target customers'; see figure 5). The use of upper- and lower-case letters of the terms used should be reviewed, as they do not always seem consistent (follow SEIR acronymous, please)

 It should be explained why the MAP of Tisanshan Town was chosen and what size it is (how many farmers involved, how many potential farmers, average size, types of relationships in the area, spread and knowledge among farmers, etc.). Lines 315-321 are not sufficient. Rows 371-376 are not clear.

A clear description and explanation of the Python model and adequate references are missing.

 The bibliography is practically only from Chinese authors. Are there no experiences outside China?

Some statements should be better explained and referred to the references (e.g. line 25: why inevitable?, line 121, information asymmetry;...). What exactly is meant by green production?

Row 371: does it mean MAP?

 The final part (row 453-462) is very unclear and should be rewritten. Also because it is not clear who the actors are who should act ('we' would be the authors?) Furthermore, these rows do not seem to be entirely justified by the results presented earlier. They should be revised.

The article does not require extensive revisions, but certainly needs much improvement in readability and homogeneity for publication.

Reviewer 2 Report

The article concerns the important issue of formation, diffusion and simulation of green production socialized service network for smallholder farmers. The chosen method seems appropriate and sufficient to investigate the problem. However, the reader has the impression that it is very synthetic, maybe even too much. Certain elements need to be completed:

- It is necessary to explain what Agricultural Green Production is - is it the production of organic food?

- what should be understood as "smallholder farmer" - how many hectares does he have? Annual turnover?

- Virtually all literature is Chinese - don't other scientists in the world deal with this topic? In a scientific article, the literature review should include different points of view on given issues, e.g. those diversified in terms of territory.

- There is no discussion of results - please complete this

- line 86- what is the abbreviation SIR model? Before using an abbreviation, the full name should be provided first, followed by the abbreviation in brackets

- line 89- what is the abbreviation SEIR model?

- line 110 – what is the abbreviation MAP?

- it is not very clear what data from the examined company was included in the study, this needs to be clarified

Best regards
